# Electrochemical Characterization of Mancozeb Degradation for Wastewater Treatment Using a Sensor Based on Poly (3,4-ethylenedioxythiophene) (PEDOT) Modified with Carbon Nanotubes and Gold Nanoparticles

**DOI:** 10.3390/polym11091449

**Published:** 2019-09-04

**Authors:** Roy Zamora-Sequeira, Fernando Alvarado-Hidalgo, Diana Robles-Chaves, Giovanni Sáenz-Arce, Esteban D. Avendano-Soto, Andrés Sánchez-Kopper, Ricardo Starbird-Perez

**Affiliations:** 1Chemistry Department, Instituto Tecnológico de Costa Rica, Cartago 30102, Costa Rica; 2Master Program in Medical Devices Engineering, Instituto Tecnológico de Costa Rica, Cartago 30102, Costa Rica; 3Chemistry Department, Centro de Investigación y de Servicios Químicos y Microbiológicos (CEQIATEC), Instituto Tecnológico de Costa Rica, Cartago 30102, Costa Rica; 4Departamento de Física, Universidad Nacional, Heredia 40101, Costa Rica; 5Physics Department, Centro de Investigación en Ciencia e Ingeniería de Materiales (CICIMA), Universidad de Costa Rica, San Pedro 11501, Costa Rica

**Keywords:** sensor, mancozeb, carbon nanotubes, electrode, conductive polymers

## Abstract

Mancozeb is a worldwide fungicide used on a large scale in agriculture. The active component and its main metabolite, ethylene thiourea, has been related to health issues. Robust, fast, and reliable methodologies to quantify its presence in water are of great importance for environmental and health reasons. The electrochemical evaluation of mancozeb using a low-cost electrochemical electrode modified with poly (3,4-ethylene dioxythiophene), multi-walled carbon nanotubes, and gold nanoparticles is a novel strategy to provide an in-situ response for water pollution from agriculture. Additionally, the thermal-, electrochemical-, and photo-degradation of mancozeb and the production of ethylene thiourea under controlled conditions were evaluated in this research. The mancozeb solutions were characterized by electrochemical oxidation and ultraviolet-visible spectrophotometry, and the ethylene thiourea concentration was measured using ultra-high-performance liquid chromatography high-resolution mass spectrometry. The degradation study of mancozeb may provide routes for treatment in wastewater treatment plants. Therefore, a low-cost electrochemical electrode was fabricated to detect mancozeb in water with a robust electrochemical response in the linear range as well as a quick response at a reduced volume. Hence, our novel modified electrode provides a potential technique to be used in environmental monitoring for pesticide detection.

## 1. Introduction 

Regions surrounded by agricultural activity have suffered from pesticide contamination. This has led to regulations and restrictions in their use. The detection of their components and degradation products is of great interest [1]. 

Mancozeb (MCZ) (Figure 1) is a dithiocarbamate used in agricultural activity [2]. MCZ acts as a biocide that inhibits the germination of spores by affecting lipid metabolism, respiration, and adenosine triphosphate (ATP) production [3]. It is used for foliar or seed treatment in banana, beans, corn, tobacco, potatoes, fruits, vegetables, and ornamentals [4]. 

MCZ is an unstable compound with low mobility because of its limited solubility in water. It is easily degraded by light, moisture, and heat [6]. In the decomposition of MCZ, the main products generated are ethylenethiourea (ETU) and ethylenebisisothiocyanate (EBIS). ETU is reported to be a compound that may affect the digestive system and the thyroid gland [7,8]. Other minor degradation products are glycine and ethylene urea (EU) [9]. Due to its high solubility in water, ETU is more toxic than MCZ and it can contaminate aquifers. Furthermore, ETU has been classified as probably carcinogenic in humans and has been associated with neurotoxic problems [10]. The degradation of some pesticides in water has been reported, but only a few studies have been found that investigate MCZ and ETU [11,12]. 

Pesticides are degraded by different mechanisms including physical, chemical, and biological degradation [13,14]. Their degradation plays a significant role in the transformation of insecticide, herbicide, and fungicide molecules into residues that are susceptible to the same forces of movement or to further degradation in the environment. Techniques for the elimination of those organic compounds from water sources include filtration [15], ozone treatment [15], electrical oxidation [16], and degradation by heat [17] and UV radiation [18,19,20]. These treatment routes in aqueous systems may offer solutions for the industrial management of contaminated water. 

The monitoring of pesticides is carried out using several analytical techniques including electrochemical sensors and liquid chromatography [21]. High-performance liquid chromatography (HPLC) coupled with multiple detection systems such as UV spectroscopy, UV-visible [22,23], and mass spectrometry [24] are the main analytical techniques used to study pesticides. These are preferred as analytical tools for their resolution power and efficiency. Separations are ultrafast with a high resolution [25].

On the other hand, electrochemical techniques have been used as counting techniques for pesticides [26] and industrial processes [27,28]. These techniques are known for being some of the most useful techniques for assessing reaction mechanisms [28] and complex electrode reactions [29]. These methods may be applied in the field. Some are non-invasive and can provide information regarding the kinetics of the sensing process [30].

Conductive polymers have been widely used in the electrochemical detection of pesticides [31,32,33]. Conductive polymers have been studied in composites with different types of functionalized nanomaterials [34,35], such as gold nanoparticles (AuNPs) [36] and carbon nanotubes (CNTs) [37], which serve as electron mediators in order to enhance the electron transfer during pesticide detection [38]. The quantity of nanoparticles on the electrode surface and their distribution may affect the sensor response [39], and force atomic microscopy is a powerful tool to evaluate these variables [40], [41].

In this study, an electrode was modified with poly (3,4-ethylenedioxythiophene) (PEDOT), multi-walled carbon nanotubes (MWCNTs), and gold nanoparticles (AuNPs) to detect MCZ. The AuNPs were characterized by particle size and ultraviolet-visible (UV-vis) spectrophotometry. In addition, thermal, ultraviolet, and electrochemical treatments were applied to mancozeb in order to evaluate the performance of the electrochemical quantification. The results were confirmed by UV spectroscopy and ultra-high-performance liquid chromatography–high-resolution mass spectrometry (UPLC-QTof-MS) systems. Therefore, our system is a suitable method for the detection of MCZ, and it might support vulnerable regions where these pesticides are applied due to agricultural activity.

## 2. Materials and Methods

### 2.1. Materials

Mancozeb 94% (*w*/*w*) (analytical standard, PESTANAL®), 2-imidazolidinethione (*w*/*w*) (analytical standard, PESTANAL®), sodium dodecyl sulfate (SDS) ACS reagent, 3,4-ethylenedioxythiophene (EDOT) (97%), and gold nanoparticles (10 nm) (product number 752584) were obtained from Sigma-Aldrich (San José, Costa Rica). Mancozeb 80% (*w*/*w*) (commercial grade: Titan 80 WP) was purchased from Dow Agro Sciences (San José, Costa Rica). Ethylenedinitrilo tetraacetic acid disodium salt (EDTA), potassium chloride (KCl), boric acid (H_3_BO_3_), phosphoric (H_3_PO_4_) acid, sodium dodecyl sulfate 99% purity, and acetic acid (CH_3_COOH) were of analytical reagent grade and were purchased from Sigma-Aldrich (San José, Costa Rica). Multi-walled carbon nanotubes (MWCNT) with an outer diameter of 5–10 nm were acquired from Cheap Tubes Inc. (Grafton, VT, USA). S1805™ Positive Photoresist and the Tetramethylammonium hydroxide (4%) UN1835 Developer were acquired from Microposit (Philadelphia, PA, USA). Polyimide DuPont™ Kapton® HN-General Purpose 12 mmil was obtained from American Durafilm (Holliston, MA, USA).

### 2.2. Mancozeb Thermal Characterization

Thermogravimetric analyses were carried out for commercial and reference Mancozeb degradation in an SDT Q600 from TA Instruments (New Castle, DE, USA) using a nitrogen atmosphere (20 mL/min) with a scan rate of 20 °C/min from ambient temperature to 900 °C in platinum sample cups (110 µL) (product code: 960149.901, TA Instruments, New Castle, DE, USA). The range of temperature was chosen for the current set based on the National Institute of Standards and Technology (NIST) SRM® 2232 (Standard Reference Materials) for calibration and validation by differential scanning calorimetry (DSC) according to the ASTM E 967. For the determination of manganese and zinc, atomic absorption equipment (Perkin Elmer, model: AAnalyst 800, Boston, MA, USA) was used. The determination of manganese and zinc in water was achieved using the methodology described in the Standard Methods for the Examination of Water (Wastewater 23rd ed. 2017 APHA-AWWA-WEF, procedures 3111-B and 3030-E), using the flame atomic absorption technique. Acid digestion with concentrated HNO_3_ at 95 °C was carried out as a preliminary preparation of the sample. The Fitness for Purpose of Analytical Methods of the EURACHEM (A Focus for Analytical Chemistry in Europe, Torino, Italy) was used to evaluate the reliability of the technique.

### 2.3. Electrode Fabrication and Characterization

Electrodes (see Figure 5b,c insets) were fabricated by the deposition of gold on a polyimide substrate using a specific shadow mask. The electrodes were passivated and etched by photolithography methods using a S1805™ photoresist (Microposit, Philadelphia, USA). 

Surfactant dispersions of SDS (CMC = 8.2 mM) were characterized by dynamic light scattering (DLS) using a Zetasizer Nano (model: ZS, Malvern, Worcestershire, UK) in order to evaluate the particle size and stability. A standard (DTS1235) was employed to confirm the proper function of the equipment. All measurements were performed in purified water at 25 °C and at a 173° angle relative to the source and sample cell types: PCS8501 and DTS1070 (Grovewood Road, UK). Additionally, the gold nanoparticle (AuNP) dispersion was studied by UV-visible spectrophotometric measurement (Thermo Scientific, model: Genesys 10S, Waltham, MA, USA) with a 10 mm quartz cell. The wavelength range was 400 to 1000 nm. Then, a 0.35% mass concentration of MWCNTs and 5 mL of 6.0 × 10^12^ particles/mL of AuNPs were added to the SDS system. Samples were sonicated before and after the addition of the monomer, EDOT (10 mM).

The electrode coating was electropolymerized from a monomer (3,4-ethylenedioxythiophene, EDOT) dispersion in a three-electrode cell configuration. The EDOT/SDS/MWCNT/AuNP solution was deposited under galvanostatic conditions using an Autolab PGSTAT 101 supplied by Metrohm (AUTOLAB, model: PGSTAT-302, Utrecht, Netherlands) with a fixed charge density of 120 mC/cm^2^ according to previous research [42]. The theoretically calculated measurements were confirmed using a 50 mM hexacyanoferrate redox standard solution.

The electrode surface topography, adhesion, and tunneling-atomic force microscopy (TUNA) electrical conductivity were obtained simultaneously using a MultiMode Nanoscope VIII atomic force microscope (AFM) Bruker operating in peak force tunneling mode (Bruker, Billerica, MA, USA). TUNA is a method for measuring currents on conductive samples. A peak force tunneling atomic force microscopy (PFTUNA) probe (with a constant force of ca. 0.4 N/m and 70 kHz of resonant frequency) was used to indent the sample surface to a depth of about 1–2 nm using a nitride cantilever. A 1 V DC bias was applied between the sample and the electrically conductive tip as the tip scanned the sample in contact mode. A linear current amplifier with a range of 50 fA to 120 pA sensed the resulting current passing through the sample. In this way, the topography, adhesion, and current image of the sample were measured simultaneously, enabling a direct correlation of the sample location with its electrical properties.

### 2.4. Preparation and Degradation of Mancozeb in Solution

#### 2.4.1. Preparation of Mancozeb Solutions 

The mancozeb samples were prepared daily by dissolving an accurately weighed amount of the compound in an EDTA solution (3%), Britton buffer, and ultrapure water at pH 7. These solutions were stored in the dark at 4 ± 1 °C until use. MCZ solutions are denoted as MCZX, where X = 0 to 100 μM, depending on the concentration.

#### 2.4.2. Thermal Degradation of Mancozeb in Solution 

A 100 μM Mancozeb solution was prepared using a reference material. For the sample analysis by thermal degradation, the solution was placed in a water bath at the following temperatures: 25, 50, 75, and 90 °C for 2 h each. For the degradation test at room temperature, a 100 μM sample was stored for six days. The performance of the modified electrodes was tested using the degradation sample at 90 ± 1 °C for 2 h.

#### 2.4.3. Photodegradation of Mancozeb in Solution under UV Light in a Photoreactor System

A 100 μM Mancozeb solution was exposed to ultraviolet radiation in a photoreactor system (Luzchem, LZC-5, Ontario, ON, Canada) with a temperature in the range of 25–30 °C using a fluorescent lamp (UVB) centered at 313 nm and placed inside the batch reactor with 16 Pyrex® test tubes topped with screw caps (approximate capacity of 10 mL). A light flux of 2.04 × 10^18^ photons/cm^2^ was calculated prior to experimentation. Aliquots were measured in triplicate every 10 min for 120 min.

The photodegradation efficiency of MCZ was calculated as follows:
(1)$$\mathrm{Photodegradation}\;\mathrm{efficiency}\;\left(\%\right)=\frac{C_{o}-C_{f}}{C_{o}}\times 100$$
where *C*_0_ is the initial concentration of pesticide and *C*_f_ is the concentration of pesticide at a certain reaction time t (min). A degradation sample exposed to UV irradiation for 120 min was used to study the electrochemical capabilities of the modified electrodes.

#### 2.4.4. Electrochemical Degradation of Mancozeb in Solution under Galvanostatic Conditions

An Autolab PGSTAT 101 (AUTOLAB, model: PGSTAT-302, Utrecht, The Netherlands) was used to apply a charge of 482.5 mC at a current of 60 μA, using platinum as working and counter electrodes, to a 100 μM mancozeb solution. A 1.5 V (vs Ag|AgCl 3 M KCl) cell potential was limited during the electrochemical degradation.

#### 2.4.5. Quantification of the Mancozeb in Solution 

The mancozeb concentrations were determined using a UV-vis spectrophotometer (Thermo Scientific, model: Genesys 10S, Waltham, MA, USA) with absorbance at 279 nm. A diverse range of performance items was subjected to instrument validation using UV Performance Validation Software in order to ensure that the UV-vis spectrophotometers offered adequate performance. Calibration curves were prepared using standard solutions of MCZ containing 0–100 μM. 

Cyclic voltammetric (CV) experiments were performed in a potentiostat (AUTOLAB, model: PGSTAT-302, Utrecht, the Netherlands) using a three-electrode configuration. The measurements were carried out using a scan rate in the range of −0.2 to 0.9 V/s for 0–100 μM MCZ solutions. A modified electrode (area: 0.1256 cm^2^) (see the inset of Figure 5b) was used as the working electrode, and the mancozeb oxidation current at 0.65 V was recorded (following a method reported in a previous research) using silver chloride (Ag|AgCl, 3.0 M KCl) as the reference electrode and gold as the counter electrode. Electrochemical impedance spectroscopy (EIS) measurements for MCZ sensing were performed with an alternating sinusoidal signal of 50 mV amplitude at open circuit potential in a potentiostat (AUTOLAB, model: PGSTAT-302, Utrecht, Netherlands) combined with a FRA32M module. Impedance spectra were collected by scanning the frequency range from 1 to 10,000 Hz. Interdigitated coated electrodes (see the inset of Figure 5c) were used as the working and counter electrodes, while an Ag|AgCl (in 3 M KCl) electrode was used as the reference electrode. The Kramers–Kronig test and the data fitting were performed using NOVA v2.1.4 software (Utrecht, the Netherlands).

A statistical measure, R-squared, was used in all calibration curves. It was calculated by systematically removing each observation from the data set and estimating the regression equation. Larger values of anticipated R^2^ suggest models of greater predictive ability (Minitab® 17.1.0., PA, USA).

In addition, ETU, a mancozeb degradation product, was measured using a Xevo G2-XS quadrupole time of flight mass (Q-tof) spectrometer (Waters Corporation, Wilmslow, UK) coupled with an Acquity UPLC H-Class. A 1 µL injection of diluted samples was separated with a ZicpHILIC Sequant column form Merck (2.1 mm × 100 mm, 5 µm) under isocratic conditions using 10% acetonitrile:water with 10 mM ammonium acetate buffer at pH 9. The mass spectrometer was configured to use a capillary voltage of 2 kV, a 10 V sampling cone, and a source offset of 90 V. Source temperatures were set at 130 and 450 °C for the desolvation temperature. For the identification, an MS mode under positive polarity was used with a mass range from 50 to 1000 m/z. Identification was based on the mass accuracy of the molecular ion (better than 1 ppm) and was accomplished by comparing the retention time with measurements of an ETU standard solution. All degradation treatments were evaluated by UV spectroscopy, cyclic voltammetry, electrochemical impedance spectroscopy, and chromatography using the calibration curves obtained previously.

## 3. Results and Discussion

### 3.1. Physicochemical Characterization of the Monomer and Nanoparticles 

Dynamic light scattering (DLS) and UV-vis spectrophotometry analysis were used to obtain the size of the different systems employed in the chemical modification of the electrode surface, as well as the commercial gold nanoparticles’ shape. The gold nanoparticles showed a maximum absorption peak in a range from 510 to 550 nm (Appendix A), which is related to the spherical shape of the gold nanoparticles [43]. Meanwhile, the results in Figure 2 show that polydispersity was found in all of the studied systems: Surfactant (SDS), SDS/EDOT, SDS/EDOT/MWCNT, and SDS/EDOT/AuNP. The SDS analysis shows two different populations, one below 10 nm that is related to SDS conglomerations and a second one between 100 and 1000 nm. This behavior is the same for of all of the systems regardless of the nature of the additive (i.e., monomer or nanoparticle).

The polymerized surface of the electrode was investigated by electrochemical AFM to obtain the surface topography, adhesion, and electrical conductivity images (see Figure 3). The topography of the PEDOT/AuNP/MWCNT coating showed height differences of around 80 nm (see Figure 3a). As for the conductivity analysis, regions of high conductivity were found regardless of the adhesion properties. This behavior might be attributed to an increase in the crystallinity characteristic of the in situ polymerization of PEDOT with CNTs via π–π interactions (see Figure 3b,c). Moreover, carbon nanotubes and gold nanoparticles contribute to the secondary charge collection that was observed in the TUNA image (see Figure 3c) [41]. In similar studies, the current differences in some places were attributed to the high mobility near the CNTs, which tend to be concentrated in specific areas [40]. Moreover, dispersed AuNPs have been reported to increase the electrical conductivity in electrochemically modified electrodes [44].

### 3.2. Thermogravimetric Characterization of Mancozeb

Thermogravimetric analysis (TGA) was performed in order to characterize both the reference and commercial MCZ in a nitrogen atmosphere. It can be seen from Figure 4a that the thermal decomposition for MCZ-C (commercial) consists of four different stages. This is very similar to the behavior previously reported [5]. Meanwhile, according to previous studies performed on the thermal decomposition of MCZ, the thermal decomposition for MCZ-R (reference) involves only three stages, as shown in Figure 4b [45].

There are three identifiable weight losses at specific temperature ranges for each stage of the decomposition process. The first stage of weight loss (WL 1) occurs in a temperature range from 100 to 198 °C, and accounted for losses of 28.89% and 16.63% for MCZ-C and MCZ-R (Figure 4), respectively. The first weight loss is associated with CS_2_ and H_2_S emissions [5,45] along with small amounts of SO_2_ and CO. H_2_S emissions are responsible for the second weight loss (WL 2) between 172 and 270 °C, corresponding to losses of 8.35 and 12.75% for MCZ-C and MCZ-R, respectively. A continuous weight loss (WL 3) was found between 269 and 832 °C with very similar weight losses for both MCZ-C and MCZ-R, 32.33 and 33.89%, respectively. WL 3 represents the last weight loss for MCZ-R. This process is attributed to the formation of inorganic ashes (i.e., MnSO_4_). However, as reported before [5] for MCZ-C, there is a fourth stage in thermal decomposition which involves the decomposition of MnSO_4_ to Mn_3_O_4_ and represents a further weight loss of 7.24%. Finally, the residues from MCZ-C and MCZ-R were 28.13 and 32.47%, respectively. These values are similar to those reported in other studies [5,45] and they match with the purity levels of both MCZ-C (80%) and MCZ-R (94%).

### 3.3. Quantification of Mancozeb in Solution 

The UV-vis spectrum for ETU, according to [46], is located in the region of 230 to 240 nm. In this range, a linear increase can be observed in Appendix A with a linear correlation of 0.97 and a value of 0.98 for the region of 265 to 285 nm. Where the linear decrease is observed, this indicates a region that other researchers have linked to MCZ [47]. The calibration curves were obtained using absorbance values at 279 nm. An R^2^ value of 0.99 was obtained with a concentration limit of 5 μM. Using linear fitting, the calibration curve had an intercept of −0.00228 ± 0.00527 arbitrary units (a.u.) of absorbance, and the value for the slope was 0.02045 ± 0.00012 a.u./µM. 

The electrical behavior of the nanoparticles observed at the nanoscale level by AFM, along with the electrochemical properties of the PEDOT [48], allowed the electrochemical quantification of MCZ. A characteristic signal for MCZ associated with the irreversible oxidation of the thiol group in the molecule [26,49] was observed close to 0.65 V in Britton Robinson buffer (pH 7). The current signal using modified electrodes was significantly larger than that obtained using bare gold electrodes (see Appendix A).

The quantification by cyclic voltammetry analysis was recorded for MCZ concentrations ranging from 0 to 100 µM. The calibration curve was calculated using current values at 0.65 V, and an R^2^ value of 0.99 was obtained with a detection limit of 5 μM. Using linear fitting, the calibration curve was shown to have an intercept of 0.25434 ± 0.01217 mA/cm^2^, and the value for the slope was 0.00413 ± 0.00025 mA/µM cm^2^. Meanwhile, Figure 5c shows Nyquist impedance plots for the mancozeb and blank solution measurements. The modified electrode electrochemical behavior and fitting has been described in previous works [48,50]. Therefore, the obtained impedance spectra were fitted to an equivalent circuit using the electrochemical model R_s_(C_DL_R_ct_)Q_1_. The solution resistance (R_s_) represents the electrical resistance of the bulk solution between the electrodes, and the capacitor component (C_DL_) is related to the double-layer capacitance. The charge transfer resistance, R_ct_, explains the charge transfer rate of the redox reaction. A Q_1_ (constant phase element) component was added, keeping an alpha value of 0.85, to describe the anomalous diffusion phenomena from the porous surface of the PEDOT/CNT/AuNP coating [51]. The value of R_ct_ increased with the mancozeb concentration, since the mechanism of recognition of the proposed sensor was based on the oxidation of the thiol groups in the mancozeb molecules, as we have previously reported [26]. In order to construct a calibration curve, the ΔR_ct_ variation was calculated by subtracting the R_ct_ of the blank solution from each MCZ solution (R_ct_) and plotting against the concentration, according to a similar study [52]. The ΔR_ct_ was found to have a linear relationship with MCZ over a concentration range of 10–100 µM/L with an R^2^ value of 0.98 and a detection limit of 5 μM. Using linear fitting, the calibration curve was found to have an intercept of −0.19 ± 0.07 Ω and the value for the slope was 0.038 ± 0.002 Ω/µM.

In order to determine the ETU, ULPC identification of the metabolite peaks in the chromatogram was achieved (see Figure 6) by a detailed evaluation of the increments and reductions of those peaks in the chromatogram at any time in comparison with the initial time. The identification of ETU in the samples was done by extracting ion chromatograms of each molecular ion sample ([M + H]^+^ of 103.0330 m/z with 1 ppm of mass error) and correlating this with the retention time for a standard signal. Confirmation of the presence of ETU in the treatments was made by electrochemical treatment, as shown in Figure 5. This result confirms the presence of ETU as one of the main metabolites of MCZ degradation, as reported by several researchers [53]. The hazard index (HI) was calculated according to the literature (SCCS, 2011. Scientific Committees on Consumer Safety, on Emerging and Newly Identified Health Risks (SCENIHR), and on Health and Environmental Risks (SCHER). The toxicity of chemical mixtures was also assessed according to the regulations of the European Commission, DG Health and Consumers. Thus, and HI above 30 for a temperature of 60 °C, and lower for UV treatments, represented an important technique for possible remediation methods.

### 3.4. Temperature and Time in the Mancozeb Degradation

The effect of temperature was studied in a 100 μM sample of MCZ prepared in Britton Robinson buffer at pH 7. The results are shown in Appendix A for samples at 25 and 90 °C. The maximum peaks of absorption for the sample at 90 °C were detected in the spectral region of 230 to 240 nm. For the sample at room temperature, the maximum peak was detected in the range of 265 to 285 nm. The spectrum found is similar to that reported in other studies for this type of fungicide [54].

### 3.5. Effect of UV Irradiation

The degradation of the fungicide by UVB irradiation was carried out on an aqueous solution of mancozeb for 2 h with sampling every 10 min. In order to determine the concentration of the fungicide, a calibration curve (R^2^ of 0.99) was used (Figure 7b). Similar studies of insecticides used in banana activity have been employed to evaluate the kinetics of UV degradation in aqueous solutions of MCZ [55]. Regarding the kinetics of UV degradation in the aqueous samples of MCZ, the equations reported in related studies were used [56]. According to the expressions and Figure 7c, it was determined that mancozeb degradation follows pseudo-first-order kinetics with a rate constant (k) of 0.0003, a half-life (t^1/2^) of about 40 min, and a ratio coefficient R^2^ of 0.98. The results of thermal degradation at a temperature of 75 °C (Figure 7) show a concentration of about 35 μM from an initial concentration of 100 μM MCZ. This is equivalent to UV degradation for 120 min with a photodegradation efficiency of 65% for MCZ (Figure 7b). Cyclic voltammetry was employed to analyze the results of UV light irradiation on 100 μM solutions of MCZ. The results are shown in Appendix A, in which a current decrease was recorded after the irradiation period. The results obtained for the photodegradation of pesticides under UV light indicate that this can be used for the treatment of mancozeb in aqueous systems in agricultural fields.

The CV technique allowed the quantification of MCZ by means of the novel PEDOT/MWCNT/AuNP electrode in a flow detection cell employing a very small volume compared to that reported in other studies [26].

### 3.6. Degradation of Mancozeb

The degradation study of mancozeb under thermal, ultraviolet, and electrochemical conditions was analyzed by ultraviolet spectroscopy and cyclic voltammetry techniques using the calibration curves (R^2^ of 0.99) obtained previously. According to the resulting data, it was observed that 74% of MCZ remains after electrochemical oxidation. The ultraviolet treatment showed that ca. 33% of MCZ withstood the UV process, and the lowest amount of MCZ (18%) was reached following the thermal analysis. The ETU was evaluated by mass chromatography for all of the samples. Only in the electrochemical oxidation treatment was ETU found at a level of 0.1 µM. Based on these results, the best treatments to be used in wastewater treatment are UV and thermal degradation, due to the absence of ETU formation during the process. However, considering the difficulty of reaching temperatures of 90 °C, it is more feasible to use ultraviolet degradation for water management.

## 4. Conclusions

The electrodes modified with the PEDOT/MWCNT/AuNP coating showed the potential to be used in the environmental quantification of mancozeb in water. The manufactured electrode demonstrated a linear behavior from 5 to 100 µM and can be considered as a useful tool for the detection of this fungicide in a short time and with a small amount of sample. The electrochemical performance was tested after applying thermal, electrical, and UV water remediation treatments to a mancozeb solution. The best treatment for mancozeb in water was found to be UV radiation, because the degradation took place without the generation of ETU. The novel sensor has potential applications in mancozeb detection as a fast detection tool for vulnerable regions and as a complementary method in water remediation studies.

## Figures and Tables

**Figure 1 polymers-11-01449-f001:**
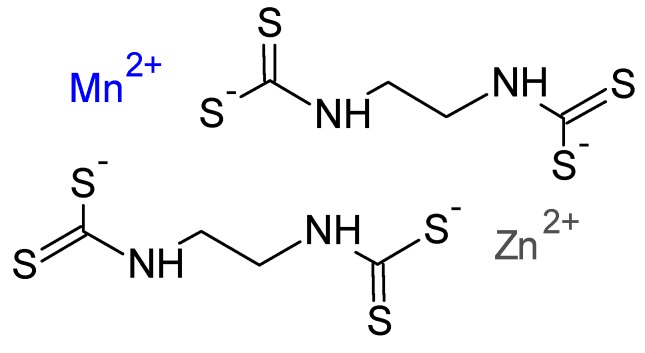
Chemical structure of mancozeb. Modified from [5].

**Figure 2 polymers-11-01449-f002:**
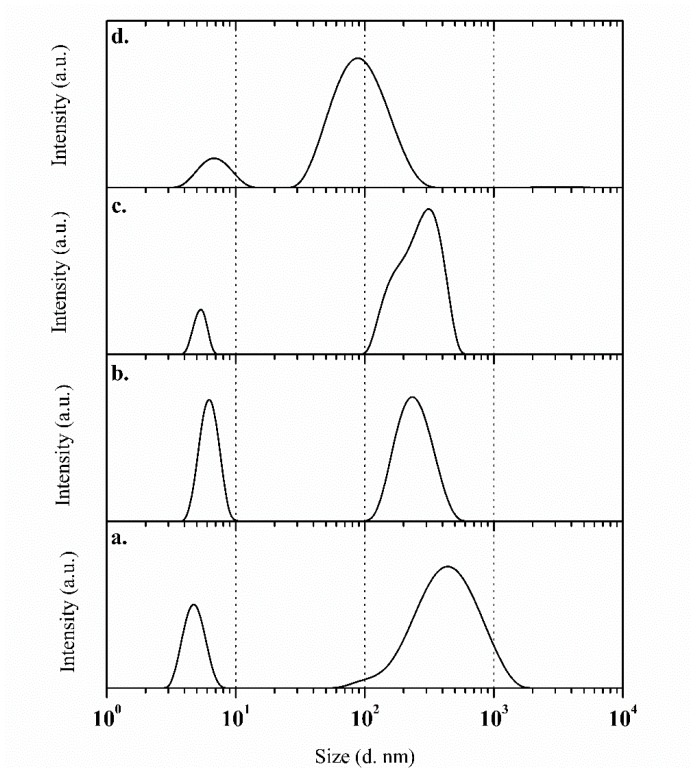
Dynamic light scattering analysis showing the size distribution of the different dispersion systems: (**a**) Sodium dodecyl sulfate (SDS), (**b**) SDS/3,4-ethylenedioxythiophene (EDOT), (**c**) SDS/EDOT/multi-walled carbon nanotube (MWCNT), and (**d**) SDS/EDOT/gold nanoparticle (AuNP) systems.

**Figure 3 polymers-11-01449-f003:**

Atomic force microscopy images of SDS/poly (3,4-ethylenedioxythiophene) (PEDOT)/AuNP/MWCNT coating, (**a**) topography, (**b**) adhesion, and (**c**) tunneling-atomic force microscopy (TUNA) electrical conductivity.

**Figure 4 polymers-11-01449-f004:**
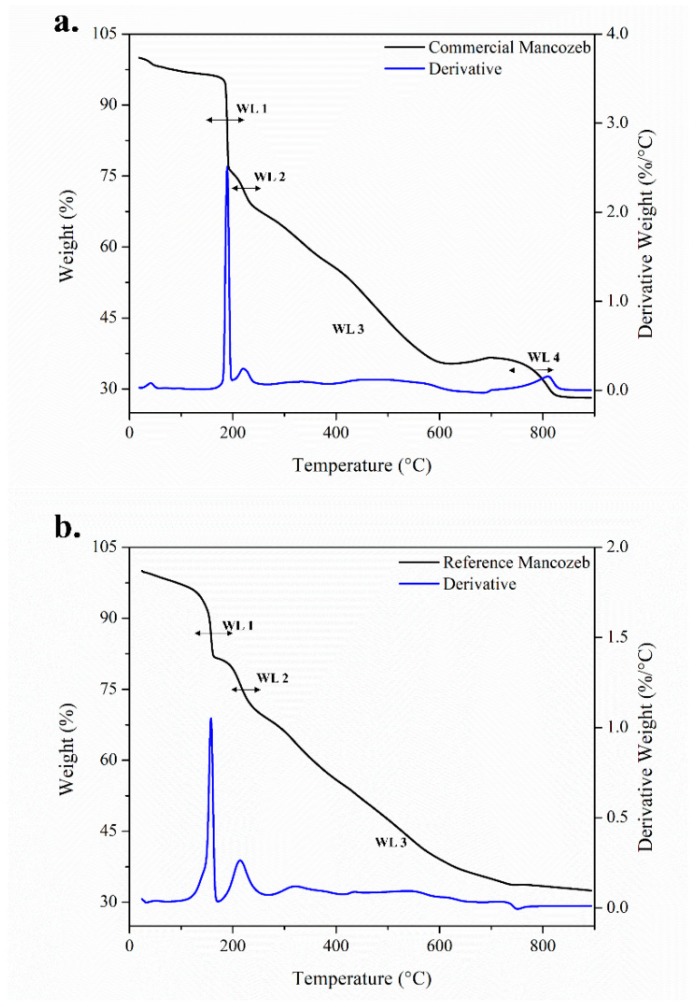
Thermograms for the mancozeb degradation profile: (**a**) Commercial presentation and (**b**) reference mancozeb.

**Figure 5 polymers-11-01449-f005:**
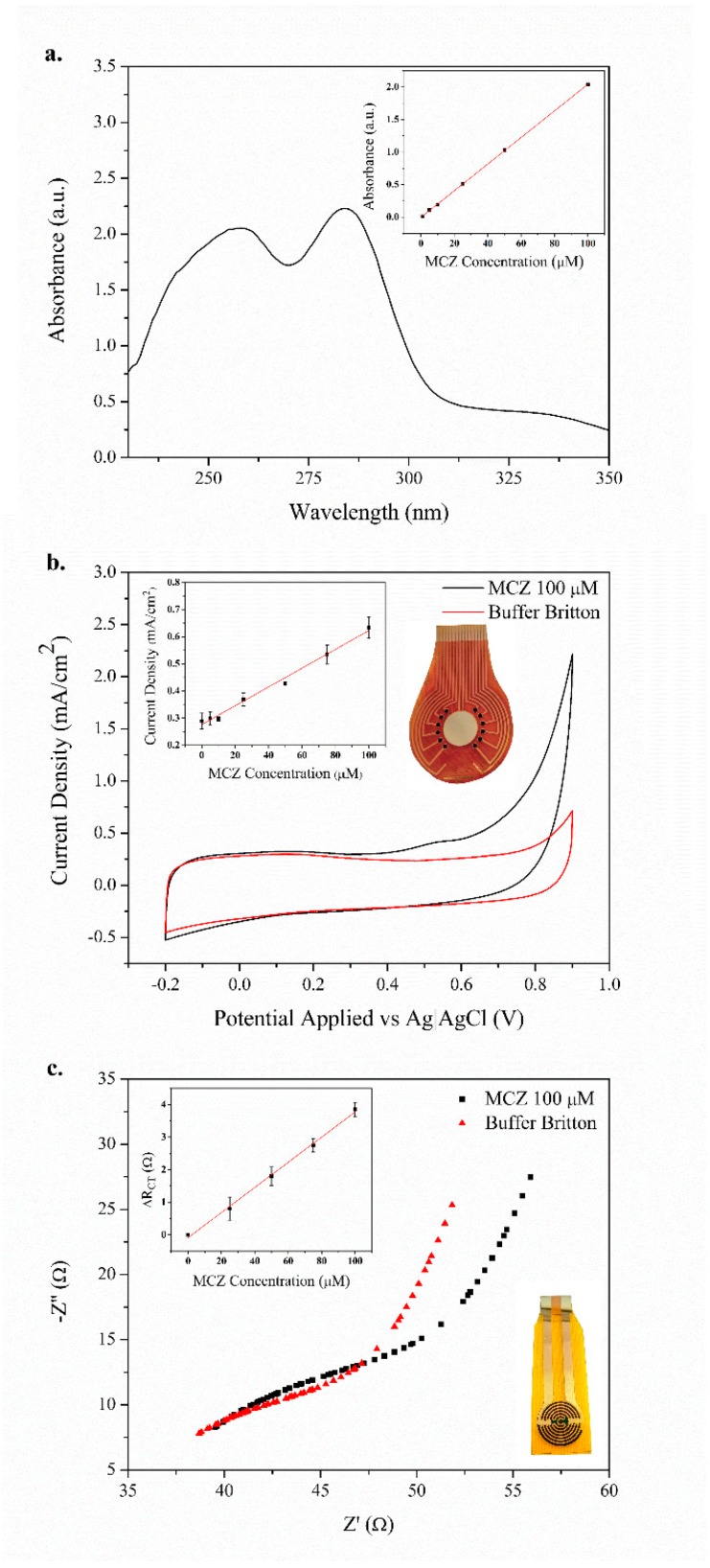
Quantification of mancozeb (inset: Calibration curves) in solution using: (**a**) UV-vis spectrophotometry, (**b**) electrochemical oxidation by cyclic voltammetry method, and (**c**) electrochemical impedance spectroscopy. Specific electrodes were fabricated regarding the technique (see insets of Figure 5b,c).

**Figure 6 polymers-11-01449-f006:**
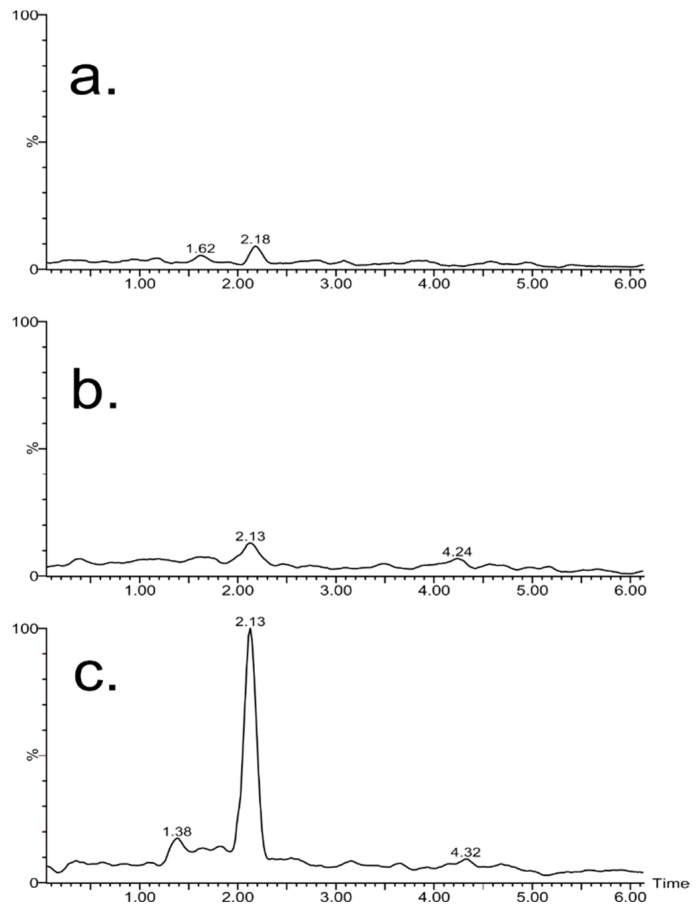
Ethylenethiourea (ETU) retention time chromatogram for each treatment sample and measured by ultra-high-performance liquid chromatography–mass spectrometry (UPLC-QTof-MS): (**a**) Thermal degradation, (**b**) UV photodegradation, and (**c**) electrochemical oxidation.

**Figure 7 polymers-11-01449-f007:**
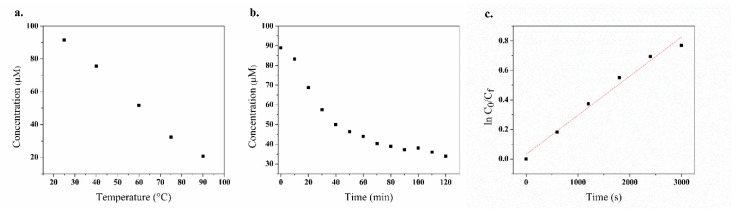
Degradation effect of temperature and UV radiation on mancozeb (MCZ). (**a**) Temperature, (**b**) time effect under UV light, and (**c**) pseudo-first-order plot for photodegradation, ln C_0_/C_f_.

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
