# Peer review of "Electrochemical Characterization of Mancozeb Degradation for Wastewater Treatment Using a Sensor Based on Poly (3,4-ethylenedioxythiophene) (PEDOT) Modified with Carbon Nanotubes and Gold Nanoparticles"

_polymers, 2019, doi:10.3390/polym11091449_

Round 1

Reviewer 1 Report

This manuscript reported a Electrochemical characterization of Mancozeb degradation for wastewater treatment using a CNT and Au sensor. This manuscript is worthy publication in Polymers, but majior revision should be made:

1. the AFM image should be replaced by another one with higher resolution. In fact, we cannot find CNT in this image.

2. The authors should  duscuss the electrochemical impedance spectroscopy in detail, and  how to ge the data on the current density vs concentration? oxidation and reduction process should be discusse in detail.

3. authors should provide the selectivity of the sensors.

4. tha data about the sensor behavior with and withou PEDOT should be provided.

5. The sensing mechanism should be disucced.

6

Author Response

Please find attached the requested document. Thank you.

Reviewer 2 Report

The manuscript present much experimental informations. However, a connection is missing between the experimental data provided. The authors could have further explored the electrochemical technique for the main focus of the study. The other experimental results could be as supplemental material. I would be in doubt about the electrochemical impedance measurements. If applied system is open circuit potential, how can redox process be explained?

What is the oxidation state of the polymer? What is the explanation for the increase in Rct with mancozeb concentration?

I do not recommend publishing the manuscript due to little scientific explanation.

Author Response

(The authors gave the same response as above.)

Reviewer 3 Report

The authors suggest investigation of the pesticide Mancozeb and some of its metabolites in conditions mimicking waste waters including potential contribution of UV irradiation in the analyte destruction. Taking into account variety of methods used and urgency of the topic of investigations related to the safe waste removal, the manuscript can be recommended for publication. The following changes and amendments are desirable:

1.      Some literature data on kinetics of degradation can be considered and compared with those obtained here

2.      Metrology aspects of the work should be described separately

3.      It is recommended to reduce first paragraphs of the Abstract and Introduction  with rather trivial information on the MCZ and products of its degradation

4.      Technical notes: refs. 32, 33 - journal titles are missed; refs. 34-36: no electroconductive polymers are used, please change references; line 15: Please give full reference description in the reference list; line 199: s-1 = ‘-1’ is a superscript; figure captions should be more informative and contain full description of the measurement conditions including concentrations, pH etc.; line 238: AFM acronym should be ascribed; lines 279-283: the number of experimental points should be added and the number of significant decimals decreased to two; line 341: dimension of the rate constant should be added.

Author Response

(The authors gave the same response as above.)

Round 2

Reviewer 1 Report

This manuscript has been revised according to the suggestion, and I suggest it can be accepted.